# SMT-based Weighted Model Integration with Structure Awareness

**Giuseppe Spallitta**[1]     **Gabriele Masina**[1]     **Paolo Morettin**[2]     **Andrea Passerini**[1]     **Roberto Sebastiani**[1]

[1]University of Trento
[2]KU Leuven

## Abstract

Weighted Model Integration (WMI) is a popular formalism aimed at unifying approaches for probabilistic inference in hybrid domains, involving logical and algebraic constraints. Despite a considerable amount of recent work, allowing WMI algorithms to scale with the complexity of the hybrid problem is still a challenge. In this paper we highlight some substantial limitations of existing state-of-the-art solutions, and develop an algorithm that combines SMT-based enumeration, an efficient technique in formal verification, with an effective encoding of the problem structure. This allows our algorithm to avoid generating redundant models, resulting in substantial computational savings. An extensive experimental evaluation on both synthetic and real-world datasets confirms the advantage of the proposed solution over existing alternatives.

## 1 INTRODUCTION

Weighted Model Integration [Belle et al., 2015] recently emerged as a unifying formalism for probabilistic inference in hybrid domains, characterized by both continuous and discrete variables and their relationships. The paradigm extends Weighted Model Counting (WMC) [Chavira and Darwiche, 2008], which is the task of computing the weighted sum of the set of satisfying assignments of a propositional formula, to deal with SMT formulas (e.g. [Barrett et al., 2009]). Whereas WMC can be made extremely efficient by leveraging component caching techniques [Sang et al., 2004, Bacchus et al., 2009], these strategies are hard to apply for WMI because of the tight coupling induced by the arithmetic constraints. Indeed, component caching approaches for WMI are restricted to fully factorized densities with few dependencies among continuous variables

[Belle et al., 2016]. Another direction specifically targets acyclic [Zeng and Van den Broeck, 2019, Zeng et al., 2020a] or loopy [Zeng et al., 2020b] pairwise models.

Exact solutions for more general classes of densities and constraints mainly leverage advancements in SMT technology or in knowledge compilation (KC) [Darwiche and Marquis, 2002]. WMI-PA [Morettin et al., 2017, 2019] relies on SMT-based Predicate Abstraction (PA) [Lahiri et al., 2006] to reduce the number of models to be generated and integrated over, and was shown to achieve substantial improvements over previous solutions. However, we show how WMI-PA has the major drawback of ignoring the structure of the weight function when pruning away redundant models. This seriously affects its simplification power when dealing with symmetries in the density. The use of KC for hybrid probabilistic inference was pioneered by Sanner and Abbasnejad [2012] and further refined in a series of later works [Kolb et al., 2018, Dos Martires et al., 2019, Kolb et al., 2020, Feldstein and Belle, 2021]. By compiling a formula into an algebraic circuit, KC techniques can exploit the structure of the problem to reduce the size of the resulting circuit, and are at the core of many state-of-the-art approaches for WMC [Chavira and Darwiche, 2008]. However, even the most recent solutions for WMI [Dos Martires et al., 2019, Kolb et al., 2020] have serious troubles in dealing with densely coupled problems, resulting in exponentially large circuits.

In this paper we introduce a novel algorithm for WMI that aims to combine the best of both worlds, by introducing weight-structure awareness into PA-based WMI. The main idea is to iteratively build a formula which mimics the conditional structure of the weight function, so as to drive the SMT-based enumeration algorithm preventing it from generating redundant models. An extensive experimental evaluation on synthetic and real-world datasets shows substantial computational advantages of the proposed solution over existing alternatives for the most challenging settings.

Our main contributions can be summarized as follows:

*Accepted for the 38th Conference on Uncertainty in Artificial Intelligence* (UAI 2022).

- We identify major efficiency issues of existing state-of-the-art WMI approaches, both PA and knowledge-compilation based.

- We introduce SA-WMI-PA, a novel WMI algorithm that combines PA with weight-structure awareness.

- We show how SA-WMI-PA achieves substantial computational improvements over existing solutions in both synthetic and real-world settings.

# 2 BACKGROUND

## 2.1 SMT AND PREDICATE ABSTRACTION

Satisfiability Modulo Theories (SMT) (see Barrett et al. [2009]) consists in deciding the satisfiability of first-order formulas over some given theory. For the context of this paper, we will refer to quantifier-free SMT formulas over linear real arithmetic ($\mathcal{LRA}$), possibly combined with uninterpreted function symbols ($\mathcal{LRA} \cup \mathcal{EUF}$). We adopt the notation and definitions in Morettin et al. [2019]. We use $\mathbb{B} \stackrel{\text{def}}{=} \{\top, \bot\}$ to indicate the set of Boolean value, whereas $\mathbb{R}$ indicates the set of real values. SMT($\mathcal{LRA}$) formulas combines Boolean variables $A_i \in \mathbb{B}$ and $\mathcal{LRA}$ atoms in the form $(\sum_i c_i x_i \bowtie c)$ (where $c_i$ are rational values, $x_i$ are real variables in $\mathbb{R}$ and $\bowtie$ is one of the standard algebraic operators $\{=, \neq, <, >, \leq, \geq\}$) by using standard Boolean operators $\{\neg, \wedge, \vee, \rightarrow, \leftrightarrow\}$. In SMT($\mathcal{LRA} \cup \mathcal{EUF}$), $\mathcal{LRA}$ terms can be interleaved with uninterpreted function symbols. Some shortcuts are provided to simplify the reading. The formula $(x_i \geq l) \wedge (x_i \leq u)$ is shortened into $[\![x_i \in [l, u]]\!]$.

Given an SMT formula $\varphi$, a *total* truth assignment $\mu$ is a function that maps *every* atom in $\varphi$ to a truth value in $\mathbb{B}$. A *partial* truth assignment maps only a subset of atoms to $\mathbb{B}$.

## 2.2 WEIGHTED MODEL INTEGRATION (WMI)

Let $\mathbf{x} \stackrel{\text{def}}{=} \{x_1, ..., x_N\} \in \mathbb{R}^N$ and $\mathbf{A} \stackrel{\text{def}}{=} \{A_1, ..., A_M\} \in \mathbb{B}^M$ for some integers $N$ and $M$. $\varphi(\mathbf{x}, \mathbf{A})$ denotes an SMT($\mathcal{LRA}$) formula over variables in $\mathbf{A}$ and $\mathbf{x}$ (subgroup of variables are admissible), while $w(\mathbf{x}, \mathbf{A})$ denotes a nonnegative weight function s.t. $\mathbb{R}^N \times \mathbb{B}^M \longmapsto \mathbb{R}^+$. Intuitively, $w$ encodes a (possibly unnormalized) density function over $\mathbf{A} \times \mathbf{x}$. Hereafter $\mu^{\mathbf{A}}$ denotes a truth assignment on $\mathbf{A}$, $\mu^{\mathcal{LRA}}$ denotes a truth assignment on the $\mathcal{LRA}$-atoms of $\varphi$, $\varphi_{[\mu^{\mathbf{A}}]}(\mathbf{x})$ denotes (any formula equivalent to) the formula obtained from $\varphi$ by substituting every Boolean value $A_i$ with its truth value in $\mu^{\mathbf{A}}$ and propagating the truth values through Boolean operators, and $w_{[\mu^{\mathbf{A}}]}(\mathbf{x}) \stackrel{\text{def}}{=} w(\mathbf{x}, \mu^{\mathbf{A}})$ is $w$ restricted to the truth values of $\mu^{\mathbf{A}}$.

Given a theory $\mathcal{T} \in \{\mathcal{LRA}, \mathcal{LRA} \cup \mathcal{EUF}\}$, the nomenclature $\mathcal{TTA}(\varphi) \stackrel{\text{def}}{=} \{\mu_1, ..., \mu_j\}$ defines the set of $\mathcal{T}$-consistent *total* assignments over both propositional and $\mathcal{T}$ atoms that

propositionally satisfy $\varphi$; $\mathcal{TA}(\varphi) \stackrel{\text{def}}{=} \{\mu_1, ..., \mu_j\}$ represents one set of $\mathcal{T}$ partial assignments over both propositional and $\mathcal{T}$ atoms that propositionally satisfy $\varphi$, s.t. every total assignment in $\mathcal{TTA}(\varphi)$ is a super-assignment of some of the partial ones in $\mathcal{TA}(\varphi)$. Given by $\varphi(\mathbf{x}, \mathbf{A})$, with $\mathcal{TTA}(\exists \mathbf{x}.\varphi)$ we mean the set of all total truth assignment $\mu^{\mathbf{A}}$ on $\mathbf{A}$ s.t. $\varphi_{[\mu^{\mathbf{A}}]}(\mathbf{x})$ is $\mathcal{T}$-satisfiable, and by $\mathcal{TA}(\exists \mathbf{x}.\varphi)$ a set of partial ones s.t. every total assignment in $\mathcal{TTA}(\exists \mathbf{x}.\varphi)$ is a super-assignment of some of the partial ones in $\mathcal{TA}(\exists \mathbf{x}.\varphi)$.

The **Weighted Model Integral** of $w(\mathbf{x}, \mathbf{A})$ over $\varphi(\mathbf{x}, \mathbf{A})$ is defined as follows [Morettin et al., 2019]:

$$\text{WMI}(\varphi, w | \mathbf{x}, \mathbf{A}) \stackrel{\text{def}}{=} \sum_{\mu^{\mathbf{A}} \in \mathbb{B}^M} \text{WMI}_{\text{nb}}(\varphi_{[\mu^{\mathbf{A}}]}, w_{[\mu^{\mathbf{A}}]} | \mathbf{x}), \quad (1)$$

$$= \sum_{\mu^{\mathbf{A}} \in \mathcal{TTA}(\exists \mathbf{x}.\varphi)} \text{WMI}_{\text{nb}}(\varphi_{[\mu^{\mathbf{A}}]}, w_{[\mu^{\mathbf{A}}]} | \mathbf{x}) \quad (2)$$

$$\text{WMI}_{\text{nb}}(\varphi, w | \mathbf{x}) \stackrel{\text{def}}{=} \int_{\varphi(\mathbf{x})} w(\mathbf{x}) \, d\mathbf{x}, \quad (3)$$

$$= \sum_{\mu^{\mathcal{LRA}} \in \mathcal{TA}(\varphi)} \int_{\mu^{\mathcal{LRA}}} w(\mathbf{x}) \, d\mathbf{x}, \quad (4)$$

where the $\mu^{\mathbf{A}}$'s are all total truth assignments on $\mathbf{A}$, $\text{WMI}_{\text{nb}}(\varphi, w | \mathbf{x})$ is the integral of $w(\mathbf{x})$ over the set $\{\mathbf{x} \mid \varphi(\mathbf{x}) \ is \ true\}$ ("nb" means "no-Booleans").

We call a *support* of a weight function $w(\mathbf{x}, \mathbf{A})$ any subset of $\mathbb{R}^N \times \mathbb{B}^M$ out of which $w(\mathbf{x}, \mathbf{A}) = 0$, and we represent it as a $\mathcal{LRA}$-formula $\chi(\mathbf{x}, \mathbf{A})$. We recall that, consequently,

$$\text{WMI}(\varphi \wedge \chi, w | \mathbf{x}, \mathbf{A}) = \text{WMI}(\varphi, w | \mathbf{x}, \mathbf{A}). \quad (5)$$

We consider the class of *feasibly integrable on $\mathcal{LRA}$* ($\text{FI}^{\mathcal{LRA}}$) functions $w(\mathbf{x})$, which contain no conditional component, and for which there exists some procedure able to compute $\text{WMI}_{\text{nb}}(\mu^{\mathcal{LRA}}, w | \mathbf{x})$ for every set of $\mathcal{LRA}$ literals on $\mathbf{x}$. (E.g., polynomials are $\text{FI}^{\mathcal{LRA}}$.) Then we call a weight function $w(\mathbf{x}, \mathbf{A})$, *feasibly integrable under $\mathcal{LRA}$ conditions* ($\text{FIUC}^{\mathcal{LRA}}$) iff it can be described in terms of a support $\mathcal{LRA}$-formula $\chi(\mathbf{x}, \mathbf{A})$ ($\top$ if not provided), a set $\Psi \stackrel{\text{def}}{=} \{\psi_i(\mathbf{x}, \mathbf{A})\}_{i=1}^K$ of $\mathcal{LRA}$ *conditions*, in such a way that, for every total truth assignment $\mu^{\Psi}$ to $\Psi$, $w_{[\mu^{\Psi}]}(\mathbf{x})$ is total and $\text{FI}^{\mathcal{LRA}}$ in the domain given by the values of $\langle \mathbf{x}, \mathbf{A} \rangle$ which satisfy $(\chi \wedge \mu^{\Psi})_{[\mu^{\mathbf{A}}]}$. $\text{FIUC}^{\mathcal{LRA}}$ functions are all the weight functions which can be described by means of arbitrary combinations of nested if-then-else's on conditions in $\mathbf{A}$ and $\Psi$, s.t. each branch $\mu^{\Psi}$ results into a $\text{FI}^{\mathcal{LRA}}$ weight function. Each $\mu^{\Psi}$ describes a portion of the domain of $w$, inside which $w_{[\mu^{\Psi}]}(\mathbf{x})$ is $\text{FI}^{\mathcal{LRA}}$, and we say that $\mu^{\Psi}$ *identifies* $w_{[\mu^{\Psi}]}$ in $w$.

In what follows we assume w.l.o.g. that $\text{FIUC}^{\mathcal{LRA}}$ functions are described as combinations of constants, variables, standard mathematical operators $+, -, \cdot, /$ un-conditioned mathematical functions (e.g., $exp, sin, ...$), conditional expressions in the form (If $\psi_i$ Then $t_{1i}$ Else $t_{2i}$) whose conditions $\psi_i$ are $\mathcal{LRA}$ formulas and terms $t_{1i}, t_{2i}$ are $\text{FIUC}^{\mathcal{LRA}}$.

**Algorithm 1** WMI-PA$(\varphi, w, \mathbf{x}, \mathbf{A})$

1: $\langle \varphi^*, w^*, \mathbf{A}^* \rangle \leftarrow \mathsf{LabelConditions}(\varphi, w, \mathbf{x}, \mathbf{A})$
2: $\mathcal{M}^{\mathbf{A}^*} \leftarrow \mathcal{TTA}(\mathsf{PredAbs}_{[\varphi^*]}(\mathbf{A}^*))$
3: $vol \leftarrow 0$
4: **for** $\mu^{\mathbf{A}^*} \in \mathcal{M}^{\mathbf{A}^*}$ **do**
5:    $\mathsf{Simplify}(\varphi^*_{[\mu^{\mathbf{A}^*}]})$
6:    **if** $\mathsf{LiteralConjunction}(\varphi^*_{[\mu^{\mathbf{A}^*}]})$ **then**
7:       $vol \leftarrow vol + \mathsf{WMI}_{\mathsf{nb}}(\varphi^*_{[\mu^{\mathbf{A}^*}]}, w^*_{[\mu^{\mathbf{A}^*}]}|\mathbf{x})$
8:    **else**
9:       $\mathcal{M}^{\mathcal{LRA}} \leftarrow \mathcal{TA}(\mathsf{PredAbs}_{[\varphi^*_{[\mu^{\mathbf{A}^*}]}]}(Atoms(\varphi^*_{[\mu^{\mathbf{A}^*}]})))$
10:       **for** $\mu^{\mathcal{LRA}} \in \mathcal{M}^{\mathcal{LRA}}$ **do**
11:          $vol \leftarrow vol + \mathsf{WMI}_{\mathsf{nb}}(\mu^{\mathcal{LRA}}, w^*_{[\mu^{\mathbf{A}^*}]}|\mathbf{x})$
12: **return** $vol$

## 2.3 WMI VIA PREDICATE ABSTRACTION

WMI-PA is an efficient WMI algorithm presented in Morettin et al. [2017, 2019] which exploits SMT-based predicate abstraction. Let $w(\mathbf{x}, \mathbf{A})$ be a $\mathsf{FIUC}^{\mathcal{LRA}}$ function as above. WMI-PA is based on the fact that

$$\mathsf{WMI}(\varphi, w|\mathbf{x}, \mathbf{A}) = \sum_{\mu^{\mathbf{A}^*} \in \mathcal{TTA}(\exists \mathbf{x}.\varphi^*)} \mathsf{WMI}_{\mathsf{nb}}(\varphi^*_{[\mu^{\mathbf{A}^*}]}, w^*_{[\mu^{\mathbf{A}^*}]}|\mathbf{x}) \quad (6)$$

$$\varphi^* \stackrel{\text{def}}{=} \varphi \wedge \chi \wedge \bigwedge_{k=1}^{K}(B_k \leftrightarrow \psi_k) \quad (7)$$

$\mathbf{A}^* \stackrel{\text{def}}{=} \mathbf{A} \cup \mathbf{B}$ s.t. $\mathbf{B} \stackrel{\text{def}}{=} \{B_1, ..., B_K\}$ are fresh propositional atoms and $w^*(\mathbf{x}, \mathbf{A} \cup \mathbf{B})$ is the weight function obtained by substituting in $w(\mathbf{x}, \mathbf{A})$ each condition $\psi_k$ with $B_k$.

The pseudocode of WMI-PA is reported in Algorithm 1. First, the problem is transformed (if needed) by labeling all conditions $\mathbf{\Psi}$ occurring in $w(\mathbf{x}, \mathbf{A})$ with fresh Boolean variables $\mathbf{B}$. After this preprocessing stage, the set $\mathcal{M}^{\mathbf{A}^*} \stackrel{\text{def}}{=} \mathcal{TTA}(\exists \mathbf{x}.\varphi^*)$ is computed by invoking SMT-based predicate abstraction Lahiri et al. [2006], namely $\mathcal{TTA}(\mathsf{PredAbs}_{[\varphi^*]}(\mathbf{A}^*))$. Then, the algorithm iterates over each Boolean assignment $\mu^{\mathbf{A}^*}$ in $\mathcal{M}^{\mathbf{A}^*}$. $\varphi^*_{[\mu^{\mathbf{A}^*}]}$ is simplified by the Simplify procedure. Then, if $\varphi^*_{[\mu^{\mathbf{A}^*}]}$ is already a conjunction of literals, the algorithm directly computes its contribution to the volume by calling $\mathsf{WMI}_{\mathsf{nb}}(\varphi^*_{[\mu^{\mathbf{A}^*}]}, w^*_{[\mu^{\mathbf{A}^*}]}|\mathbf{x})$. Otherwise, $\mathcal{TA}(\varphi^*_{[\mu^{\mathbf{A}^*}]})$ is computed as $\mathcal{TA}(\mathsf{PredAbs}_{[\varphi^*_{[\mu^{\mathbf{A}^*}]}]}(Atoms(\varphi^*_{[\mu^{\mathbf{A}^*}]})))$ to produce partial assignments, and the algorithm iteratively computes contributions to the volume for each $\mu^{\mathcal{LRA}}$. We refer the reader to Morettin et al. [2019] for more details.

Notice that in the actual implementation the potentially-large sets $\mathcal{M}^{\mathbf{A}^*}$ and $\mathcal{M}^{\mathcal{LRA}}$ are not generated explicitly. Rather, their elements are generated, integrated and then dropped one-by-one, so that to avoid space blowup.

## 3 EFFICIENCY ISSUES

### 3.1 KNOWLEDGE COMPILATION

We start our analysis of WMI techniques by noticing a major problem with existing KC approaches for WMI [Dos Martires et al., 2019, Kolb et al., 2020], in that they tend easily to blow up in space even with simple weight functions. Consider, e.g., the case in which

$$w(\mathbf{x}, \mathbf{A}) \stackrel{\text{def}}{=} \prod_{i=1}^{N}(\text{if } \psi_i \text{ then } w_{i1}(\mathbf{x}) \text{ else } w_{i2}(\mathbf{x})) \quad (8)$$

where the $\psi_i$s are $\mathcal{LRA}$ conditions on $\{\mathbf{x}, \mathbf{A}\}$ and the $w_{i1}, w_{i2}$ are generic functions on $\mathbf{x}$. First, the decision diagrams do not interleave arithmetical and conditional operators, rather they push all the arithmetic operators below the conditional ones. Thus with (8) the resulting decision diagrams consist of $2^N$ branches on the $\psi_i$s, each corresponding to a distinct unconditioned weight function of the form $\prod_{i=1}^{N} w_{ij_i}(\mathbf{x})$ s.t. $j_i \in \{1, 2\}$. Second, the decision diagrams are built on the Boolean abstraction of $w(\mathbf{x}, \mathbf{A})$, s.t. they do not eliminate a priori the useless branches consisting in $\mathcal{LRA}$-inconsistent combinations of $\psi_i$s, which can be up to exponentially many.

With WMI-PA, instead, the representation of (8) does not grow in size, because $\mathsf{FIUC}^{\mathcal{LRA}}$ functions allow for interleaving arithmetical and conditional operators. Also, the SMT-based enumeration algorithm does not generate $\mathcal{LRA}$-inconsistent assignments on the $\psi_i$s. We stress the fact that (8) is not an artificial scenario: rather, e.g., this is the case of the real-world logistics problems in Morettin et al. [2019].

### 3.2 WMI-PA

We continue our analysis by noticing a major deficiency also of the WMI-PA algorithm, that is, it fails to leverage the structure of the weight function to prune the set of models to integrate over. We illustrate the issue by means of a simple example (see Figure 1).

**Example 1** *Let $\varphi \stackrel{\text{def}}{=} \top$, $\chi \stackrel{\text{def}}{=} [\![x_1 \in [0, 2]]\!] \wedge [\![x_2 \in [0, 3]]\!]$ (Figure 1(a)) and let $w(\mathbf{x}, \mathbf{A})$ be a tree-structured weight function defined as in Figure 1(b). To compute $\mathsf{WMI}(\varphi \wedge \chi, w|\mathbf{x}, \mathbf{A})$, six integrals have to be computed:*

*$x_1^2 x_2$ on $[\![x_1 \in [1, 2]]\!] \wedge [\![x_2 \in [1, 3]]\!]$ (if $A_1 = \top$)*
*$x_1^3 x_2$ on $[\![x_1 \in [1, 2]]\!] \wedge [\![x_2 \in [0, 1]]\!]$ (if $A_1 = \top$)*
*$x_1 x_2^2$ on $[\![x_1 \in [0, 1]]\!] \wedge [\![x_2 \in [2, 3]]\!]$ (if $A_1 = \top$)*
*$x_1 x_2^3$ on $[\![x_1 \in [0, 1]]\!] \wedge [\![x_2 \in [0, 2]]\!]$ (if $A_1 = \top$).*
*$2x_1 x_2$ on $[\![x_1 \in [0, 2]]\!] \wedge [\![x_2 \in [0, 3]]\!]$ (if $A_1 = \bot, A_2 = \top$)*
*$3x_1 x_2$ on $[\![x_1 \in [0, 2]]\!] \wedge [\![x_2 \in [0, 3]]\!]$ (if $A_1 = \bot, A_2 = \bot$)*

*When WMI-PA is used (Algorithm 1), applying LabelCon-*

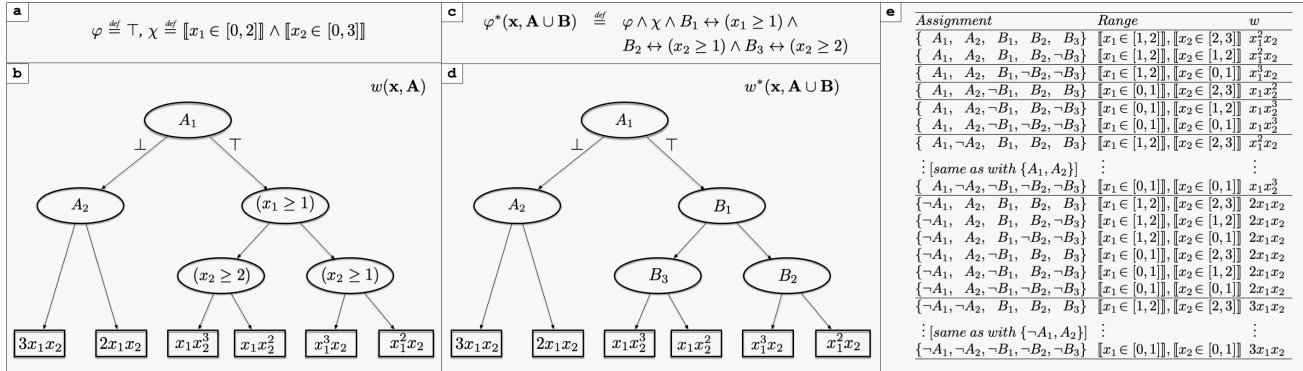

Figure 1: Example highlighting the efficiency issues of the WMI-PA algorithm. (**a**) definition of formula $\varphi$ (trivially true) and support $\chi$. (**b**) definition of the weight function $w(\mathbf{x}, \mathbf{A})$. Round nodes indicate if-then-else conditions, with true and false cases on the right and left outgoing edges respectively. Squared nodes indicate $\mathsf{FI}^{\mathcal{LRA}}$ weight functions. (**c**) novel version of the formula $\varphi^*(\mathbf{x}, \mathbf{A} \cup \mathbf{B})$ after the application of the LabelConditions(...) step of WMI-PA. (**d**) novel version of the weight function $w^*(\mathbf{x}, \mathbf{A} \cup \mathbf{B})$, where all $\mathcal{LRA}$ conditions have been replaced with the fresh Boolean variables introduced in $\varphi^*(\mathbf{x}, \mathbf{A} \cup \mathbf{B})$. (**e**) List of assignments obtained by WMI-PA on $\mathbf{A} \cup \mathbf{B}$. Notice the amount of assignments sharing the same $\mathsf{FI}^{\mathcal{LRA}}$ weight function.

ditions(...) *we obtain (Figure 1(c)):*

$$\varphi^*(\mathbf{x}, \mathbf{A} \cup \mathbf{B}) \quad \stackrel{\text{def}}{=} \quad \varphi \wedge \chi \wedge B_1 \leftrightarrow (x_1 \geq 1) \wedge$$
$$B_2 \leftrightarrow (x_2 \geq 1) \wedge B_3 \leftrightarrow (x_2 \geq 2)$$

*and the weight function $w^*(\mathbf{x}, \mathbf{A} \cup \mathbf{B})$ shown in Figure 1(d). Then, by applying $\mathcal{TTA}(\mathsf{PredAbs}_{[\varphi^*]}(\mathbf{A}^*))$ (row 2) we obtain 24 total assignments $\mathcal{M}^{\mathbf{A}^*}$ on $\mathbf{A} \cup \mathbf{B}$, as shown in Figure 1(e).. Notice that WMI-PA uselessly splits into 2 parts the integrals on $x_1^2 x_2$ and $x_1 x_2^3$ and into 6 parts the integral on $2x_1 x_2$ and on $3x_1 x_2$. Also, it repeats the very same integrals for $\{A_1, A_2, ...\}$ and $\{A_1, \neg A_2, ...\}$.* $\diamond$

We highlight two facts. First, WMI-PA enumerates *total* truth assignments on the Boolean atoms $\mathbf{A} \cup \mathbf{B}$ in $\mathcal{TTA}(\exists \mathbf{x}.\varphi^*)$ (6) (row 2 in Algorithm 1), assigning also unnecessary values. Second, WMI-PA labels $\mathcal{LRA}$ conditions in $w$ by means of fresh Boolean atoms $\mathbf{B}$ (row 1 in Algorithm 1). This forces the enumerator to assign all their values in every assignment, even when not necessary.

The key issue about WMI-PA is that the enumeration of $\mathcal{TTA}(\exists \mathbf{x}.\varphi^*)$ in (6) and of $\mathcal{TA}(\varphi^*_{[\mu^{\mathbf{A}^*}]})$ in (4) (rows 2 and 11 in Algorithm 1) *is not aware of the conditional structure of the weight function $w$*, in particular, it is not aware of the fact that often *partial* assignments to the set of conditions in $w^*$ (both Boolean and $\mathcal{LRA}$) are sufficient to identify the value of a $\mathsf{FIUC}^{\mathcal{LRA}}$ function (e.g $\{A_1, B_1, B_2\}$ suffices to identify $x_1^2 x_2$, or $\{\neg A_1, A_2\}$ suffices to identify $2x_1 x_2$), so that it is forced to enumerate all total assignments extending them (e.g. $\{A_1, A_2, B_1, B_2, B_3\}$ and $\{A_1, A_2, B_1, B_2, \neg B_3\}$).

Thus, to cope with this issue, we need to modify WMI-PA to make it aware of the conditional structure of $w$.

## 4 MAKING WMI-PA WEIGHT-STRUCTURE AWARE

The key idea to prevent total enumeration works as follows. We do *not* rename with $\mathbf{B}$ the conditions $\mathbf{\Psi}$ in $w$ and, rather, than enumerating total truth assignments for $\exists \mathbf{x}.\varphi^*$ as in (6)–(7), we enumerate *partial* assignments for $\exists \mathbf{x}.\varphi^{**}$ where $\varphi^{**} \stackrel{\text{def}}{=} \varphi \wedge \chi \wedge \mathsf{sk}(w)$ and $\mathsf{sk}(w)$ –which we call the *conditional skeleton* of $w$– is a $\mathcal{LRA}$ formula s.t.:
(*a*) its atoms are all and only the conditions in $\mathbf{\Psi}$,
(*b*) is $\mathcal{LRA}$-valid, so that $\varphi \wedge \chi$ is equivalent to $\varphi \wedge \chi \wedge \mathsf{sk}(w)$,
(*c*) any *partial* truth value assignment $\mu$ to the conditions $\mathbf{\Psi}$ which makes $\mathsf{sk}(w)$ true is such that $w_{[\mu]}$ is $\mathsf{FI}^{\mathcal{LRA}}$.[1]
Thus, we have that (2) can be rewritten as:

$$\mathsf{WMI}(\varphi, w|\mathbf{x}, \mathbf{A}) = \sum_{\mu \in \mathcal{TA}(\exists \mathbf{x}.\varphi^{**})} 2^{|\mathbf{A} \setminus \mu|} \cdot \mathsf{WMI}_{\mathsf{nb}}(\varphi^{**}_{[\mu]}, w_{[\mu]}|\mathbf{x}) \quad (9)$$

$$\varphi^{**} \quad \stackrel{\text{def}}{=} \quad \varphi \wedge \chi \wedge \mathsf{sk}(w) \quad (10)$$

where $|\mathbf{A} \setminus \mu|$ is the number of Boolean atoms in $\mathbf{A}$ that are not assigned by $\mu$. Condition (*c*) guarantees that $\mathsf{WMI}_{\mathsf{nb}}(\varphi^{**}_{[\mu]}, w_{[\mu]}|\mathbf{x})$ in (9) can be directly computed, without further partitioning. The $2^{|\mathbf{A} \setminus \mu|}$ factor in (9) resembles the fact that, if some Boolean atom $A_i \in \mathbf{A}$ is not assigned in $\mu$, then $\mathsf{WMI}_{\mathsf{nb}}(\varphi^{**}_{[\mu]}, w_{[\mu]}|\mathbf{x})$ should be counted twice because $\mu$ represents two assignments $\mu \cup \{A_i\}$ and $\mu \cup \{\neg A_i\}$ which would produce two identical integrals.

Notice that logic-wise $\mathsf{sk}(w)$ is non-informative because it is a valid formula. Nevertheless, the role of $\mathsf{sk}(w)$ is to mimic the structure of $w$ so that to "make the enumerator aware of the presence of the conditions $\mathbf{\Psi}$", forcing every

---

[1] E.g., the partial assignment $\mu \stackrel{\text{def}}{=} \{A_1, (x_1 \geq 1), (x_2 \geq 1)\}$ in Example 1 is such that $w_{[\mu]} = x_1^2 x_2$, which is $\mathsf{FI}^{\mathcal{LRA}}$.

assignment $\mu$ to assign truth values also to these conditions which are necessary to make $w_{[\mu]}$ FI$^{\mathcal{LRA}}$ and hence make WMI$_{\mathsf{nb}}(\varphi_{[\mu]}^{**}, w_{[\mu]}|\mathbf{x})$ directly computable, without further partitioning.

An important issue is to avoid $\mathsf{sk}(w)$ blow up in size. E.g., one could use as $\mathsf{sk}(w)$ a formula encoding the conditional structure of an XADDs or (F)XSDDs, but this may cause a blow up in size, as discussed in Section 3.1.

In order to prevent such problems, we do not generate $\mathsf{sk}(w)$ explicitly. Rather, we build it as a disjunction of partial assignments over $\mathbf{\Psi}$ which we enumerate progressively. To this extent, we define $\mathsf{sk}(w) \stackrel{\text{def}}{=} \exists \mathbf{y}.[\![y = w]\!]$ where $[\![y = w]\!]$ is a formula on $\mathbf{A}, \mathbf{x}, \mathbf{y}$ s.t. $\mathbf{y} \stackrel{\text{def}}{=} \{y, y_1, ..., y_k\}$ is a set of fresh variables. Thus, $\mathcal{TA}(\exists \mathbf{x}.(\varphi \wedge \chi \wedge \exists \mathbf{y}.[\![y = w]\!]))$ can be computed as $\mathcal{TA}(\exists \mathbf{xy}.(\varphi \wedge \chi \wedge [\![y = w]\!]))$ because the $\mathbf{y}$'s do not occur in $\varphi \wedge \chi$, with no need to generate $\mathsf{sk}(w)$ explicitly. The enumeration of $\mathcal{TA}(\exists \mathbf{xy}.(\varphi \wedge \chi \wedge [\![y = w]\!]))$ is performed by the very same SMT-based procedure used in Morettin et al. [2019].

$[\![y = w]\!]$ is obtained by taking $(y = w)$, s.t. $y$ is fresh, and recursively substituting bottom-up every conditional term (If $\psi_i$ Then $t_{i1}$ Else $t_{i2}$) in it with a fresh variable $y_i \in \mathbf{y}$, adding the definition of $(y_i = (\text{If } \psi_i \text{ Then } t_{i1} \text{ Else } t_{i2}))$ as

$$(\neg\psi_i \vee y_i = t_{i1}) \wedge (\psi_i \vee y_i = t_{i2}). \quad (11)$$

This labeling&rewriting process, which is inspired to labeling CNF-ization [Tseitin, 1968], guarantees that the size of $[\![y = w]\!]$ is linear wrt. that of $w$. E.g., if (8) holds, then $[\![y = w]\!]$ is $(y = \prod_{i=1}^{N} y_i) \wedge \bigwedge_{i=1}^{N}((\neg\psi_i \vee y_i = w_{i1}(\mathbf{x})) \wedge (\psi_i \vee y_i = w_{i2}(\mathbf{x})))$.

One problem with the above definition of $[\![y = w]\!]$ is that it is not a $\mathcal{LRA}$-formula, because $w$ may include multiplications or even transcendental functions out of the conditions $\mathbf{\Psi}^2$, which makes SMT reasoning over it dramatically hard or even undecidable. We notice, however, that when computing $\mathcal{TA}(\exists \mathbf{xy}.(\varphi \wedge \chi \wedge [\![y = w]\!]))$ the arithmetical functions (including operators $+, -, *, /$) occurring in $w$ out of the conditions $\mathbf{\Psi}$ have no role, since the only fact that we need to guarantee for the validity of $\mathsf{sk}(w)$ is that they are indeed functions, so that $\exists y.(y = f(...))$ is always valid.[3] (In substance, during the enumeration we are interested only in the truth values of the conditions $\mathbf{\Psi}$ in $\mu$ which make $w_{[\mu]}$ FI$^{\mathcal{LRA}}$, regardless the actual values of $w_{[\mu]}$). Therefore we can safely substitute condition-less arithmetical functions (including operators $+, -, \cdot, /$) with some fresh uninterpreted function symbols, obtaining a $\mathcal{LRA} \cup \mathcal{EUF}$-formula $[\![y = w]\!]_{\mathcal{EUF}}$, which is relatively easy to solve by standard SMT solvers [Barrett et al., 2009]. It is easy to see

that a partial assignment $\mu$ evaluating $[\![y = w]\!]$ to true is $\mathcal{LRA}$-satisfiable iff its corresponding assignment $\mu_{\mathcal{EUF}}$ is $\mathcal{LRA} \cup \mathcal{EUF}$-satisfiable.[4] Therefore, we can modify the enumeration procedure into $\mathcal{TA}(\exists \mathbf{xy}.(\varphi \wedge \chi \wedge [\![y = w]\!]_{\mathcal{EUF}}))$.

Finally, we enforce the fact that the two branches of an if-then-else are alternative by adding to (11) a mutual-exclusion constraint $\neg(y_i = t_{i1}) \vee \neg(y_i = t_{i2})$, so that the choice of the function is univocally associated to the list of decisions on the $\psi_i$s. The procedure producing $[\![y = w]\!]_{\mathcal{EUF}}$ is described in detail in Appendix, Algorithm 1.

**Example 2** *Consider the problem in Example 1. Figure 2 shows the relabeling process applied to the weight function $w$. The resulting $[\![y = w]\!]_{\mathcal{EUF}}$ formula is:*

$$[\![y = w]\!](\mathbf{x} \cup \mathbf{y}, \mathbf{A}) \stackrel{\text{def}}{=}$$
$$(\neg A_1 \vee \neg(x_1 \geq 1) \vee \neg(x_2 \geq 1) \qquad \vee \ (y_1 = f_{11}(\mathbf{x})))$$
$$\wedge(\neg A_1 \vee \neg(x_1 \geq 1) \vee \ (x_2 \geq 1) \qquad \vee \ (y_1 = f_{12}(\mathbf{x})))$$
$$\wedge(\neg A_1 \vee \neg(x_1 \geq 1) \vee \neg(y_1 = f_{11}(\mathbf{x})) \vee \neg(y_1 = f_{12}(\mathbf{x})))$$
$$\wedge(\neg A_1 \vee \ (x_1 \geq 1) \vee \neg(x_2 \geq 2) \qquad \vee \ (y_2 = f_{21}(\mathbf{x})))$$
$$\wedge(\neg A_1 \vee \ (x_1 \geq 1) \vee \ (x_2 \geq 2) \qquad \vee \ (y_2 = f_{22}(\mathbf{x})))$$
$$\wedge(\neg A_1 \vee \ (x_1 \geq 1) \vee \neg(y_2 = f_{21}(\mathbf{x})) \vee \neg(y_2 = f_{22}(\mathbf{x})))$$
$$\wedge(\neg A_1 \vee \neg(x_1 \geq 1) \qquad\qquad \vee \ (y_3 = y_1))$$
$$\wedge(\neg A_1 \vee \ (x_1 \geq 1) \qquad\qquad \vee \ (y_3 = y_2))$$
$$\wedge(\neg A_1 \vee \neg(y_3 = y_1) \vee \neg(y_3 = y_2))$$
$$\wedge(\ A_1 \vee \neg A_2 \qquad\qquad \vee \ (y_4 = f_3(\mathbf{x})))$$
$$\wedge(\ A_1 \vee \ A_2 \qquad\qquad \vee \ (y_4 = f_4(\mathbf{x})))$$
$$\wedge(\ A_1 \vee \neg(y_4 = f_3(\mathbf{x})) \vee \neg(y_4 = f_4(\mathbf{x})))$$
$$\wedge(\neg A_1 \qquad\qquad\qquad \vee \ (y_5 = y_3))$$
$$\wedge(\ A_1 \qquad\qquad\qquad \vee \ (y_5 = y_4))$$
$$\wedge(\neg(y_5 = y_3) \vee \neg(y_5 = y_4))$$
$$\wedge(\ (y = y_5))$$

*Figure 3 illustrates a possible enumeration process. The algorithm enumerates partial assignments satisfying $\varphi \wedge \chi \wedge [\![y = w]\!]_{\mathcal{EUF}}$, restricted on the conditions $\mathbf{\Psi} \stackrel{\text{def}}{=} \{A_1, A_2, (x_1 \geq 1), (x_2 \geq 1), (x_2 \geq 2)\}$, which is equivalent to enumerate $\mathcal{TA}(\exists \mathbf{xy}.(\varphi \wedge \chi \wedge [\![y = w]\!]_{\mathcal{EUF}}))$. Assuming the enumeration procedure picks nondeterministic choices following the order of the above set[5] and assigning positive values first, then in the first branch the following satisfying partial assignment is generated, in order:[6]*
$$\chi \cup \{(y = y_5), \underline{A_1}, (y_5 = y_3), \neg(y_5 = y_4), \underline{(x_1 \geq 1)},$$
$$(y_3 = y_1), \neg(y_3 = y_2), \underline{(x_2 \geq 1)}, (y_1 = f_{11}(\mathbf{x})),$$
$$\neg(y_1 = f_{12}(\mathbf{x}))\}.$$

*(Notice that, following the chains of true equalities, we have $y = y_5 = y_3 = y_1 = f_{11}(\mathbf{x})$.) Then the SMT*

---

[2]The conditions in $\mathbf{\Psi}$ contain only linear terms by definition.

[3]This propagates down on the recursive structure of $[\![y = w]\!]$ because, if $y$ does not occur in $\psi$, $\exists y.(y = (\text{If } \psi \text{ Then } t_1 \text{ Else } t_2))$ is equivalent to $((\text{If } \psi \text{ Then } \exists y.(y = t_1) \text{ Else } \exists y.(y = t_2)))$, and $\varphi$ is equivalent to $\exists y.(\varphi[t|y] \wedge (y = t))$.

[4]This boils down to the fact that $y$ occurs only in the top equation and as such it is free to assume arbitrary values, and that all arithmetic functions are total in the domain so that, for every possible values of $\mathbf{x}$ a value for $\mathbf{y}$ always exists iff there exists in the $\mathcal{EUF}$ version.

[5]Like in Algorithm 2, we pick Boolean conditions first.

[6]Here nondeterministic choices are underlined. The atoms in $\chi$ are assigned deterministically.

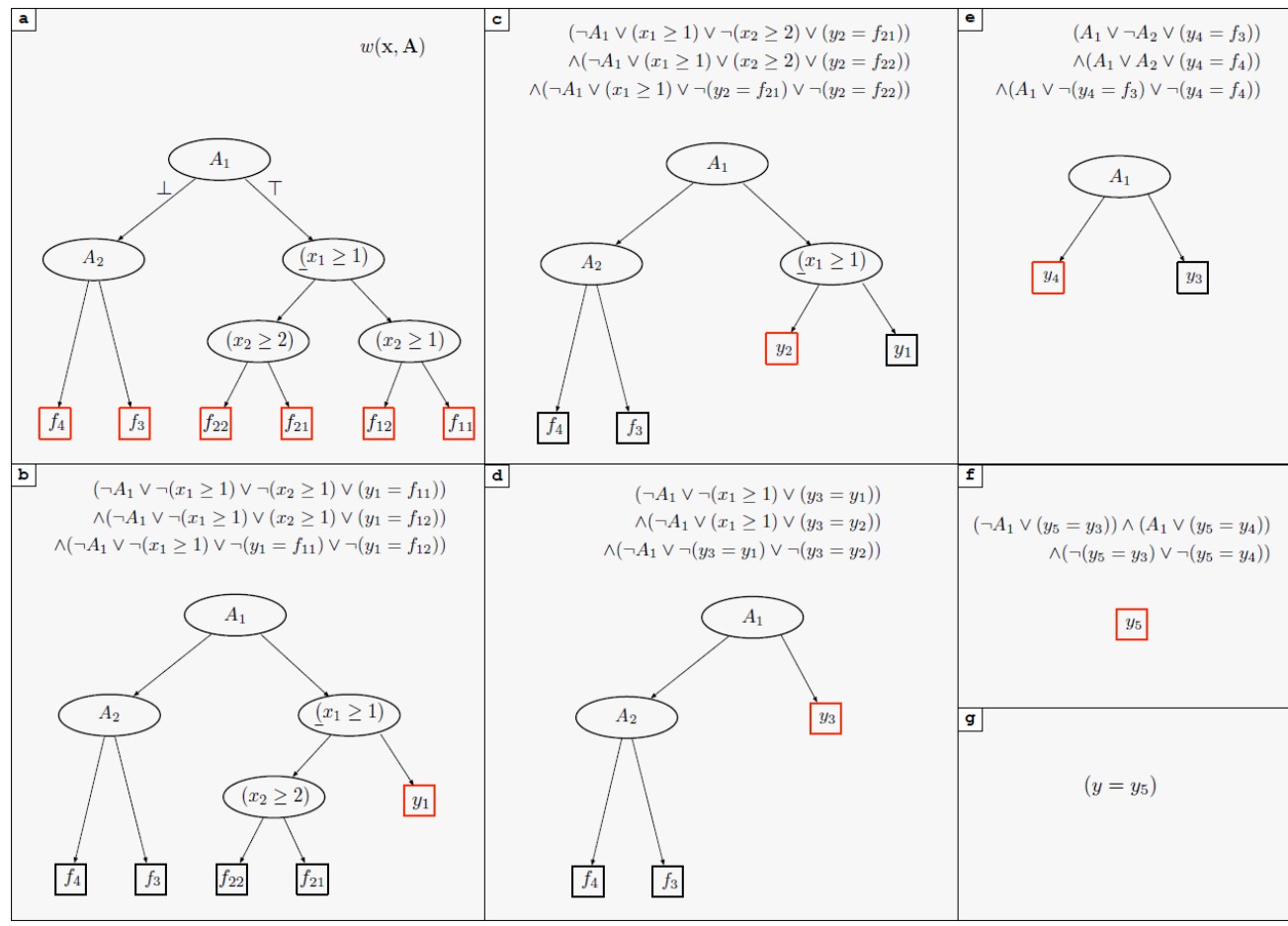

Figure 2: Example of bottom-up procedure for computing the relabeling function $[\![y = w]\!]_{\mathcal{E}\mathcal{U}\mathcal{F}}$. **(a)** Replacement of $\mathsf{FI}^{\mathcal{L}\mathcal{R}\mathcal{A}}$ weight functions (the leaves of the tree, highlighted in red) with $\mathcal{E}\mathcal{U}\mathcal{F}$ function symbols (we dropped the dependency on $x$ for compactness). **(b-g)** Sequence of relabeling steps. At each step, a conditional term is replaced by a fresh $\mathcal{L}\mathcal{R}\mathcal{A}$ variable $y_i$. The encoding of the variable in shown in the upper part, while the lower part shows the weight function with the branch of the conditional term replaced with $y_i$ (highlighted in red). The last step consists in renaming the top variable as $y$, so that $y = w(\mathbf{x}, \mathbf{A})$. The relabeling function $[\![y = w]\!]_{\mathcal{E}\mathcal{U}\mathcal{F}}$ is simply the conjunction of the encodings in the different steps.

*solver extracts from it the subset $\{A_1, (x_1 \geq 1), (x_2 \geq 1)\}$ restricted on the conditions in $\Psi$. Then the blocking clause $\neg A_1 \vee \neg(x_1 \geq 1) \vee \neg(x_2 \geq 1)$ is added to the formula, which prevents to enumerate the same subset again. This forces the algorithm to backtrack and generate*
$$\chi \cup \{(y = y_5), \underline{A_1}, (y_5 = y_3), \neg(y_5 = y_4), (x_1 \geq 1),$$
$$(y_3 = y_1), \neg(y_3 = y_2), \neg(x_2 \geq 1), (y_1 = f_{12}(\mathbf{x})),$$
$$\neg(y_1 = f_{11}(\mathbf{x}))\}.$$
*producing the assignment:* $\{A_1, (x_1 \geq 1), \neg(x_2 \geq 1)\}.$ [7]

*Overall, the algorithm enumerates the following ordered collection of partial assignments restricted to $Atoms(\varphi \wedge \chi) \cup \mathbf{\Psi}$:*

$$\chi \cup \{\ A_1,\quad (x_1 \geq 1),\quad (x_2 \geq 1)\},\quad //y = ... = f_{11}(\mathbf{x})$$
$$\chi \cup \{\ A_1,\quad (x_1 \geq 1), \neg(x_2 \geq 1)\},\quad //y = ... = f_{12}(\mathbf{x})$$
$$\chi \cup \{\ A_1, \neg(x_1 \geq 1),\quad (x_2 \geq 2)\},\quad //y = ... = f_{21}(\mathbf{x})$$
$$\chi \cup \{\ A_1, \neg(x_1 \geq 1), \neg(x_2 \geq 2)\},\quad //y = ... = f_{22}(\mathbf{x})$$
$$\chi \cup \{\neg A_1,\quad A_2\},\quad //y = ... = f_3(\mathbf{x})$$
$$\chi \cup \{\neg A_1, \neg A_2\}\quad //y = ... = f_4(\mathbf{x})$$

*which correspond to the six integrals of Example 1. Notice that according to (2) the first four integrals have to be multiplied by 2, because the partial assignment $\{A_1\}$ covers two total assignments $\{A_1, A_2\}$ and $\{A_1, \neg A_2\}$. Notice also that the disjunction of the six partial assignments,*
$$(A_1 \wedge (x_1 \geq 1) \wedge (x_2 \geq 1)) \vee ... \vee (\neg A_1 \wedge \neg A_2),\ matches\ the$$
*definition of $\mathsf{sk}(w)$, which we have computed by progressive enumeration rather than encoded a priori.* ◇

---

[7]We refer the reader to Lahiri et al. [2006] for more details on the SMT-based enumeration algorithm.

Based on the previous ideas, we develop SA-WMI-PA, a novel "weight structure aware" variant of WMI-PA. The

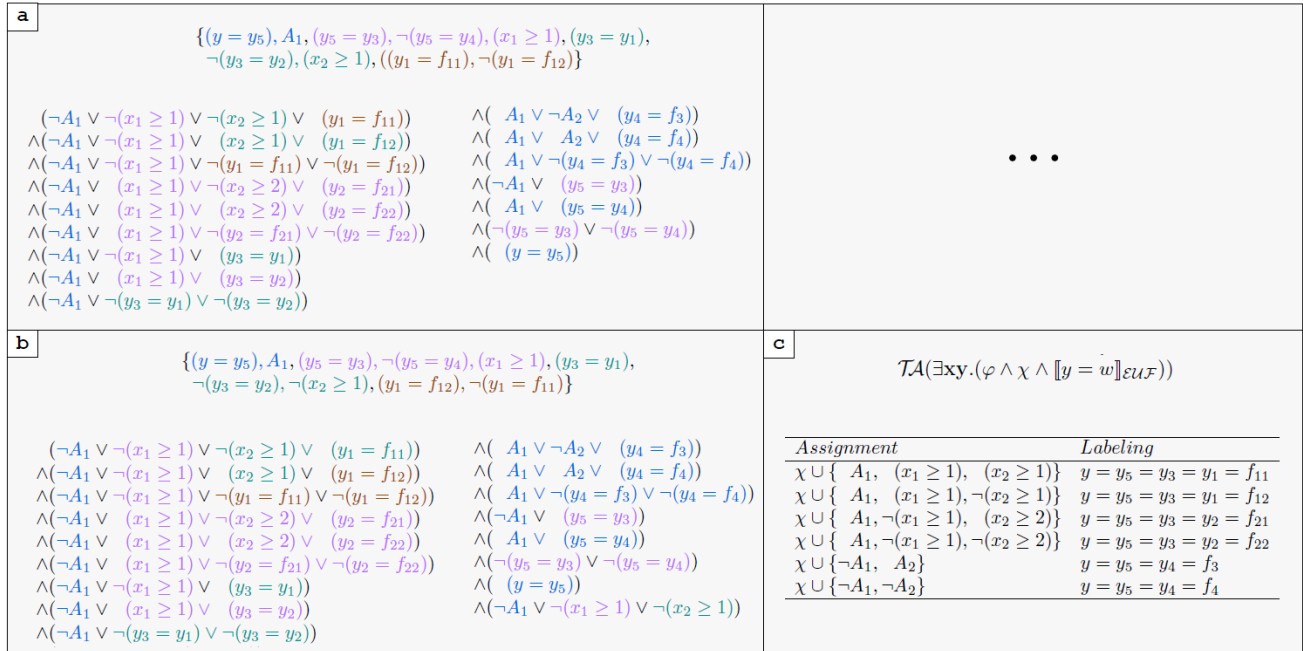

Figure 3: Example of structure-aware enumeration performed by SA-WMI-PA on the problem in Example 1. **(a)** Generation of the first assignment. The assignment is on top, while the bottom part shows the $[\![y = w]\!]_{\mathcal{E}\mathcal{U}\mathcal{F}}$ formula. Colors indicate the progression of the generation, in terms of atoms added (top) and parts of the formula to be removed as a consequence (bottom). For the sake of simplicity all atoms until the next atom in $\Psi \overset{def}{=} \{A_1, A_2, (x_1 \geq 1), (x_2 \geq 1), (x_2 \geq 2)\}$ (if any) are given the same color. **(b)** Generation of the second assignment. Note how the $[\![y = w]\!]_{\mathcal{E}\mathcal{U}\mathcal{F}}$ formula is enriched with the blocking clause $\neg A_1 \vee \neg(x_1 \geq 1) \vee \neg(x_2 \geq 1)$ preventing the first assignment to be generated again. **(c)** Final result of the enumeration (which contains six assignments in total). The partial assignments are obtained by restricting the generated assignments on the conditions in $\Psi$ (and combining them with the atoms of $\chi$, which here are assigned deterministically.). For each assignment, the corresponding chain of equivalences of the **y**s with the identified leaf $\mathsf{FI}^{\mathcal{L}\mathcal{R}\mathcal{A}}$ function is displayed.

pseudocode of SA-WMI-PA is reported in Algorithm 2.

As with WMI-PA, we enumerate the assignments in two main steps: in the first loop (rows 3-7) we generate a set $\mathcal{M}^{\mathbf{A}^*}$ of partial assignments $\mu^{\mathbf{A}^*}$ over the Boolean variables $\mathbf{A}$, s.t. $\varphi^{**}_{[\mu^{\mathbf{A}}]}$ is $\mathcal{L}\mathcal{R}\mathcal{A}$-satisfiable and does not contain Boolean variables anymore. In Example 2 $\mathcal{M}^{\mathbf{A}^*} \overset{def}{=} \{\{A_1\}, \{\neg A_1, A_2\}, \{\neg A_1, \neg A_2\}\}$. In the second loop (rows 11-19), for each $\mu^{\mathbf{A}^*}$ in $\mathcal{M}^{\mathbf{A}^*}$ we enumerate the set $\mathcal{M}^{\mathcal{L}\mathcal{R}\mathcal{A}}$ of $\mathcal{L}\mathcal{R}\mathcal{A}$-satisfiable partial assignments satisfying $\varphi^{**}_{[\mu^{\mathbf{A}^*}]}$ (that is, on $\mathcal{L}\mathcal{R}\mathcal{A}$ atoms in $Atoms(\varphi \wedge \chi) \cup \Psi$), we compute the integral $\mathsf{WMI}_{\mathsf{nb}}(\mu^{\mathcal{L}\mathcal{R}\mathcal{A}}, w_{[\mu^{\mathbf{A}*}]}|\mathbf{x})$, multiply it by the $2^{|\mathbf{A} \setminus \mu^{\mathbf{A}^*}|}$ factor and add it to the result. In Example 2, if e.g. $\mu^{\mathbf{A}^*} = \{A_1\}$, $\mathcal{T}\mathcal{A}(\mathsf{PredAbs}_{[\varphi^{**}_{[\mu^{\mathbf{A}^*}]}]}(Atoms(\varphi^{**}_{[\mu^{\mathbf{A}^*}]})))$ computes the four partial assignments $\{\chi \cup \{(x_1 \geq 1), (x_2 \geq 1)\}, ..., \chi \cup \{\neg(x_1 \geq 1), \neg(x_2 \geq 2)\}\}$.

In detail, in row 2 we extend $\varphi \wedge \chi$ with $[\![y = w]\!]_{\mathcal{E}\mathcal{U}\mathcal{F}}$ to provide structure awareness. (We recall that, unlike with WMI-PA, we do not label $\mathcal{L}\mathcal{R}\mathcal{A}$ conditions with fresh Boolean variables $\mathbf{B}$.) Next, in row 3 we perform $\mathcal{T}\mathcal{A}(\mathsf{PredAbs}_{[\varphi^{**}_{[\mu^{\mathbf{A}}]}]}(\mathbf{A}))$ to obtain a set $\mathcal{M}^{\mathbf{A}}$ of partial as-

signments restricted on Boolean atoms $\mathbf{A}$. Then, for each assignment $\mu^{\mathbf{A}} \in \mathcal{M}^{\mathbf{A}}$ we build the (simplified) residual $\varphi^{**}_{[\mu^{\mathbf{A}}]}$. Since $\mu^{\mathbf{A}}$ is partial, $\varphi^{**}_{[\mu^{\mathbf{A}}]}$ is not guaranteed to be free of Boolean variables $\mathbf{A}$, as shown in Example 3. If this is the case, we simply add $\mu^{\mathbf{A}}$ to $\mathcal{M}^{\mathbf{A}^*}$, otherwise we invoke $\mathcal{T}\mathcal{T}\mathcal{A}(\mathsf{PredAbs}_{[\varphi^{**}_{[\mu^{\mathbf{A}}]}]}(\mathbf{A}))$ to assign the remaining variables and conjoin each assignment $\mu^{\mathbf{A}}_{residual}$ to $\mu^{\mathbf{A}}$, ensuring that the residual now contains only $\mathcal{L}\mathcal{R}\mathcal{A}$ atoms (rows 4-10). The second loop (rows 11-19) resembles the main loop in WMI-PA, with the only relevant difference that, since $\mu^{\mathbf{A}^*}$ is partial, the integral is multiplied by a $2^{|\mathbf{A} \setminus \mu^{\mathbf{A}^*}|}$ factor.

Notice that in general the assignments $\mu^{\mathbf{A}^*}$ are partial even if the steps in rows 9-10 are executed; the set of residual Boolean variables in $\varphi^{**}_{[\mu^{\mathbf{A}}]}$ are a (possibly much smaller) subset of $\mathbf{A} \setminus \mu^{\mathbf{A}}$ because some of them do not occur anymore in $\varphi^{**}_{[\mu^{\mathbf{A}}]}$ after the simplification, as shown in the following example.

**Example 3** *Let* $\varphi \overset{def}{=} (A_1 \vee A_2 \vee A_3) \wedge (\neg A_1 \vee A_2 \vee (x \geq 1)) \wedge (\neg A_2 \vee (x \geq 2)) \wedge (\neg A_3 \vee (x \leq 3))$, $\chi \overset{def}{=} [\![x_1 \in [0,4]]\!]$ *and* $w(\mathbf{x}, \mathbf{A}) \overset{def}{=} 1.0$. *Suppose* $\mathcal{T}\mathcal{A}(\mathsf{PredAbs}_{[\varphi^{**}]}(\mathbf{A}))$ *finds the partial assignment*

**Algorithm 2** SA-WMI-PA($\varphi, w, \mathbf{x}, \mathbf{A}$)

1: $\mathcal{M}^{\mathbf{A}^*} \leftarrow \emptyset; vol \leftarrow 0$
2: $\varphi^{**} \leftarrow \varphi \wedge \chi \wedge [\![y = w]\!]_{\mathcal{EUF}}$
3: $\mathcal{M}^{\mathbf{A}} \leftarrow \mathcal{TA}(\text{PredAbs}_{[\varphi^{**}]}(\mathbf{A}))$
4: **for** $\mu^{\mathbf{A}} \in \mathcal{M}^{\mathbf{A}}$ **do**
5: $\quad$ Simplify($\varphi^{**}_{[\mu^{\mathbf{A}}]}$)
6: $\quad$ **if** $\varphi^{**}_{[\mu^{\mathbf{A}}]}$ does not contain Boolean variables **then**
7: $\quad\quad$ $\mathcal{M}^{\mathbf{A}^*} \leftarrow \mathcal{M}^{\mathbf{A}^*} \cup \{\mu^{\mathbf{A}}\}$
8: $\quad$ **else**
9: $\quad\quad$ **for** $\mu^{\mathbf{A}}_{residual} \in \mathcal{TTA}(\text{PredAbs}_{[\varphi^{**}_{[\mu^{\mathbf{A}}]}]}(\mathbf{A}))$ **do**
10: $\quad\quad\quad$ $\mathcal{M}^{\mathbf{A}^*} \leftarrow \mathcal{M}^{\mathbf{A}^*} \cup \{\mu^{\mathbf{A}} \wedge \mu^{\mathbf{A}}_{residual}\}$
11: **for** $\mu^{\mathbf{A}^*} \in \mathcal{M}^{\mathbf{A}^*}$ **do**
12: $\quad$ $k \leftarrow |\mathbf{A} \setminus \mu^{\mathbf{A}^*}|$
13: $\quad$ Simplify($\varphi^{**}_{[\mu^{\mathbf{A}^*}]}$)
14: $\quad$ **if** LiteralConjunction($\varphi^{**}_{[\mu^{\mathbf{A}^*}]}$) **then**
15: $\quad\quad$ $vol \leftarrow vol + 2^k \cdot \text{WMI}_{\text{nb}}(\varphi^{**}_{[\mu^{\mathbf{A}^*}]}, w_{[\mu^{\mathbf{A}^*}]}|\mathbf{x})$
16: $\quad$ **else**
17: $\quad\quad$ $\mathcal{M}^{\mathcal{LRA}} \leftarrow \mathcal{TA}(\text{PredAbs}_{[\varphi^{**}_{[\mu^{\mathbf{A}^*}]}]}(Atoms(\varphi^{**}_{[\mu^{\mathbf{A}^*}]})))$
18: $\quad\quad$ **for** $\mu^{\mathcal{LRA}} \in \mathcal{M}^{\mathcal{LRA}}$ **do**
19: $\quad\quad\quad$ $vol \leftarrow vol + 2^k \cdot \text{WMI}_{\text{nb}}(\mu^{\mathcal{LRA}}, w_{[\mu^{\mathbf{A}^*}]}|\mathbf{x})$
20: **return** $vol$

---

$\{(x \geq 0), (x \leq 4), A_2, (x \geq 1), (x \geq 2), (x \leq 3)\}$, *whose projected version is* $\mu^{\mathbf{A}} \stackrel{def}{=} \{A_2\}$ *(row 3). Then* $\varphi^{**}_{[\mu^{\mathbf{A}}]}$ *reduces to* $(\neg A_3 \vee (x \leq 3))$, *so that* $\mathcal{M}^{\mathbf{A}^*}$ *is* $\{\{A_2, A_3\}, \{A_2, \neg A_3\}\}$, *avoiding branching on* $A_1$. $\quad\diamond$

We stress the fact that in our actual implementation, like with that of WMI-PA, the potentially-large sets $\mathcal{M}^{\mathbf{A}^*}$ and $\mathcal{M}^{\mathcal{LRA}}$ are not generated explicitly. Rather, their elements are generated, integrated and then dropped one-by-one, so that to avoid space blowup.

We highlight two main differences wrt. WMI-PA. First, unlike with WMI-PA, the generated assignments $\mu^{\mathbf{A}}$ on $\mathbf{A}$ are partial, each representing $2^{|\mathbf{A} \setminus \mu^{\mathbf{A}}|}$ total ones. Second, the assignments on (non-Boolean) conditions $\mathbf{\Psi}$ inside the $\mu^{\mathcal{LRA}}$s are also partial, whereas with WMI-PA the assignments to the $\mathbf{B}$s are total. This may drastically reduce the number of integrals to compute, as empirically demonstrated in the next section.

# 5 EXPERIMENTAL EVALUATION

The novel algorithm SA-WMI-PA is compared to the original WMI-PA algorithm [Morettin et al., 2019], and the WMI solvers based on KC: XADD [Kolb et al., 2018], XSDD and FXSDD [Kolb et al., 2020]. Each of these methods is called from the Python framework pywmi [Kolb et al., 2019]. For both WMI-PA and SA-WMI-PA, we use MATHSAT for SMT enumeration and LATTE INTE-

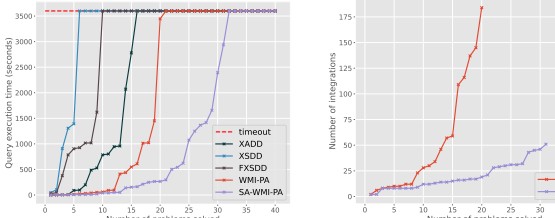

Figure 4: Cactus plots reporting execution time for all methods on the synthetic experiments (left); number of integrals (right) for WMI-PA and SA-WMI-PA.

GRALE for computing integrals. For the KC algorithms we use PSiPSI [Gehr et al., 2016] as symbolic computer algebra backend. All experiments are performed on an Intel Xeon Gold 6238R @ 2.20GHz 28 Core machine with 128GB of ram and running Ubuntu Linux 20.04. The code of SA-WMI-PA is freely available at https://github.com/unitn-sml/wmi-pa.

For improved readability, in both the experiments we report runtime using cactus plots, i.e. the single problem instances are increasingly sorted by runtime for each algorithm separately. We highlight how, by construction, problem instances of the same tick of the x-axis are not guaranteed to be the same for different algorithms. Steeper slopes of an algorithm curve means less efficiency.

## 5.1 SYNTHETIC EXPERIMENTS

We first evaluate our algorithm on random formulas and weights, following the experimental protocol of Morettin et al. [2019]. We define two recursive procedures to generate $\mathcal{LRA}$ formulae and weight functions with respect to a positive integer number $D$, named depth:

$$r_\varphi(D) = \begin{cases} \bigoplus_{q=1}^Q r_\varphi(D-1) & \text{if } D > 0 \\ [\neg]A^{\mathbb{B}/\mathbb{R}} & \text{otherwise} \end{cases}$$

$$r_w(D) = \begin{cases} \begin{cases} [\![\text{If } r_\varphi(D) \text{ Then } r_w(D-1) \text{ Else } r_w(D-1)]\!] \\ \text{or} \\ r_w(D-1) \otimes r_w(D-1) \end{cases} & \text{if } D > 0 \\ P_{\text{random}}(\mathbf{x}) & \text{otherwise} \end{cases}$$

where $\bigoplus \in \{\bigvee, \bigwedge, \neg\bigvee, \neg\bigwedge\}, \otimes \in \{+, \cdot\}$ and $P_{\text{random}}(\mathbf{x})$ is a random polynomial function. Using these procedure we generate instances of synthetic problems:

$$\chi(\mathbf{x}, \mathbf{A}) = r_\varphi(D) \wedge \bigwedge_{x \in \mathbf{x}} [\![x \in [l_x, u_x]]\!]$$
$$w(\mathbf{x}, \mathbf{A}) = r_w(D)$$
$$\varphi_{\text{query}}(\mathbf{x}, \mathbf{A}) = r_\varphi(D)$$

where $l_x, u_x$ are real numbers such that $\forall x.(l_x < u_x)$.

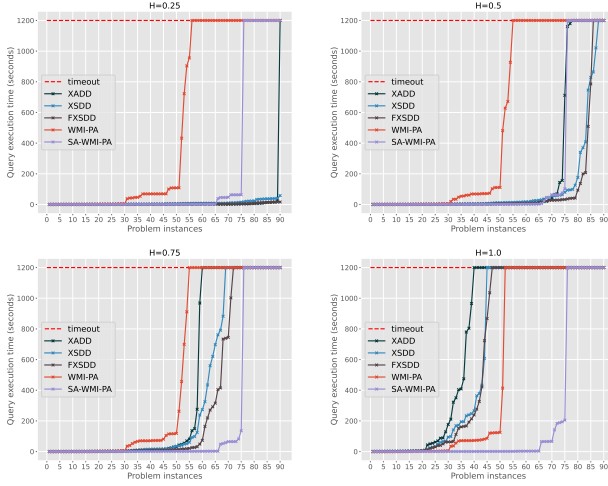

Figure 5: Cactus plots representing average query execution times and standard deviation in seconds on a set of DET problems with $H \in \{0.25, 0.5, 0.75, 1\}$.

In contrast with the benchmarks used in other recent works [Kolb et al., 2018, 2020], the procedure is not strongly biased towards the generation of problems with structural regularities, offering a more neutral perspective on how the different techniques are expected to perform in the wild. The generated synthetic benchmark contains problems where the number of both Boolean and real variables is set to 3, while the depth of weights functions fits in the range $[4, 7]$. Timeout was set to 3600 seconds, similarly to what has been done in previous works.

In this settings, the approaches based on a SMT oracle clearly outperform those based on KC (Fig. 4 left). In addition, SA-WMI-PA greatly improves over WMI-PA, thanks to a drastic reduction in the number of integrals computed (Fig. 4 right), with the advantage of our approach getting more evident when the weight functions are deeper.

### 5.2 DENSITY ESTIMATION TREES

We explore the use of WMI solvers for marginal inference in real-world probabilistic models. In particular, we considered *Density Estimation Trees* (DETs) [Ram and Gray, 2011], hybrid density estimators encoding piecewise constant distributions. Having only univariate conditions in the internal nodes, DETs natively support tractable inference when single variables are considered. Answering queries like $Pr(X \leq Y)$ requires instead marginalizing over an oblique constraint, which is reduced to WMI: $Pr(X \leq Y) = \frac{WMI((X \leq Y) \wedge \chi_{DET}, w_{DET})}{WMI(\chi_{DET}, w_{DET})}$. Being able to address this type of queries is crucial to apply WMI-based inference to e.g. probabilistic formal verification tasks, involving constraints that the system should satisfy with high probability.

We considered a selection of hybrid datasets from the UCI repository [Dua and Graff, 2017], reported in Table 1 in the Appendix. Following the approach of Morettin et al. [2020], discrete numerical features were relaxed into continuous variables, while $n$-ary categorical features are one-hot encoded with $n$ binary variables.

After learning a DET on each dataset, we generated a benchmark of increasingly complex queries, 5 for each dataset, involving a ratio $H \in [0, 1]$ of the continuous variables. More specifically, the queries are linear inequalities involving a number of variables $k = max(1, \lfloor H \cdot |\mathbf{x}| \rfloor)$. Figure 5 depicts the runtime of the algorithms for $H \in \{0.25, 0.5, 0.75, 1\}$. Timeout was set to 1200 seconds. KC approaches have an edge for the simplest cases ($H \leq 0.5$) in which substantial factorization of the integrals is possible. Contrarily to many other probabilistic models, which are akin to the case in Section 3.1, DETs are well-suited for KC-based inference, due to the absence of arithmetic operations in the internal nodes. When the coupling between variables increases, however, the advantage of decomposing the integrals is overweight by the combinatorial reasoning capabilities of SA-WMI-PA. We remark that SA-WMI-PA is agnostic of the underlying integration procedure, and thus in principle it could also incorporate a symbolic integration component.

## 6 CONCLUSION

We presented the first SMT-based algorithm for WMI that is aware of the structure of the weight function. This is particularly beneficial when the piecewise density defined on top of $SMT(\mathcal{LRA})$ constraints is deep, as it is often the case with densities learned from data. We evaluated our algorithmic ideas on both synthetic and real-world problems, obtaining state-of-the-art results in many settings of practical interest. Providing unprecedented scalability in the evaluation of complex probabilistic SMT problems, this contribution directly impacts the use of WMI for the probabilistic verification of systems. While the improvements described in this work drastically reduce the number of integrations required to compute a weighted model integral, the integration itself remains an obstacle to scalability. Combining the fast combinatorial reasoning offered by SMT solvers with symbolic integration is a promising research direction.

### Acknowledgements

This research was partially supported by TAILOR, a project funded by EU Horizon 2020 research and innovation programme under GA No 952215 and from the European Research Council (ERC) under the European Union's Horizon 2020 research and innovation programme (grant agreement No. [694980] SYNTH: Synthesising Inductive Data Models). The authors would like to thank Pedro Zuidberg Dos Martires for his help with XSDDs and FXSDDs.

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
