# OpenReview forum: "SMT-based Weighted Model Integration with Structure Awareness"
_auai.org/UAI/2022/Conference — UAI 2022 Poster_

### Official Review · Reviewer_rJjv · 2022-04-11

**Q2(1) Originality/Novelty:** 3
**Q2(2) Significance/Impact:** 3
**Q2(3) Correctness/Technical Quality:** 3
**Q2(6) Clarity Of Writing:** 2
**Q6 Overall Score:** 6
**Q8 Confidence In Your Score:** 3

**Q1 Summary And Contributions:**

This paper proposes an algorithm for the Weighted Model Integration (WMI) problem. The paper fist discusses performance issues with recent approaches to WMI: WMI with predicate abstraction (WMI-PA) and via knowledge compilation (KC). The paper proposes a novel algorithm, SA-WMI-PA with certain structure awareness. An implementation of the proposed algorithm is empirically evaluated against WMI-PA and KC methods.

**Q2 Assessment Of The Paper:**

More detailed information regarding each of these aspects is given below:

**Q2(4) Quality Of Experiments (Optional):**

3: Good: The experimental evaluation is adequate, and the results convincingly support the main claims.

**Q2(5) Reproducibility:**

3: Good: Key resources (e.g., proofs, code, data) are available and key details (e.g., proofs, experimental setup) are sufficiently well-described for competent researchers to confidently reproduce the main results.

**Q3 Main Strengths:**

I think that the main strength of the paper is the proposal of a novel algorithm for WMI (dubbed SA-WMI-PA) which is subsequently empirically evaluated, and showing promise on certain instance families (compared to other methods). A main part of the paper is on performance issues of previous methods/algorithms, which is also helpful.

**Q4 Main Weakness:**

While understandable to some extend for conference versions, I think the paper could be improved regarding its accessibility. The definition of WMI is, naturally, somewhat involved, and some more explanations and examples would help in more quickly understanding the formalism. However, I would note that the discussion around shortcomings of previous approaches also helps in understanding WMI.

The experimental evaluation is interesting, but it seems to include not many instances. Judging from Figure 4 and Figure 5, the number of instances considered is less than 100? If there is a misunderstanding, I'd ask the authors to respond. Nevertheless, the experiments give an interesting indication of performance of the novel algorithm. The size of some instances also appears to be not very large (at least for the synthetic instances).

Nevertheless, it seems to me that the experiments show potential usefulness of the proposed algorithm.

**Q5 Detailed Comments To The Authors:**

I think that the present paper tackles a relevant problem (WMI), and provides a potentially useful algorithmic solution that avoids certain redundancies present in earlier approaches. The presentation is adequate, but could be improved.

**Q7 Justification For Your Score:**

Overall, I view the paper in a positive light. The main assessment of the paper is based on the experimental evaluation. There is a bit of incrementality in the proposed algorithm, since it is based on an earlier ones. Nevertheless, the experimental evaluation suggest strength of the proposed algorithm. Before response and discussion, my recommendation is that of weak accept. In the current manuscript, I do not see a clear argument to raise my evaluation.

**Q9 Complying With Reviewing Instructions:**

1: Yes.

---

### Official Review · Reviewer_7JHv · 2022-04-13

**Q2(1) Originality/Novelty:** 3
**Q2(2) Significance/Impact:** 3
**Q2(3) Correctness/Technical Quality:** 3
**Q2(6) Clarity Of Writing:** 2
**Q6 Overall Score:** 7
**Q8 Confidence In Your Score:** 2

**Q1 Summary And Contributions:**

An approach is proposed to make weighted model integration more efficient. The main contribution is an approach that shows how we could use structure in the weight function to reduce the number of possible enumerations.

**Q2 Assessment Of The Paper:**

More detailed information regarding each of these aspects is given below:

**Q2(4) Quality Of Experiments (Optional):**

3: Good: The experimental evaluation is adequate, and the results convincingly support the main claims.

**Q2(5) Reproducibility:**

3: Good: Key resources (e.g., proofs, code, data) are available and key details (e.g., proofs, experimental setup) are sufficiently well-described for competent researchers to confidently reproduce the main results.

**Q3 Main Strengths:**

Overall the paper solves a problem that seems significant since WMI is widely applicable.
The idea of using the weight function to guide the SMT enumeration seems to be novel.

**Q4 Main Weakness:**

The writing could perhaps be improved to show more intuitive examples. Some of the examples in the figures were hard to understand.

**Q5 Detailed Comments To The Authors:**

An approach is proposed to make weighted model integration more efficient. Typically, in hybrid domains we need to perform many integrations, this approach reduces this by taking advantage of problem structure. Specifically, the model is represented as SMT formulas and there can be a mix of logical and arithmetic constraints specified. Typically SMT-based enumeration can blow up in size. The idea here is to use structure to reduce this and improve scalability. Particularly, when the weight function has structure, we may not need the full enumeration. Experiments are performed on synthetic problems and also on marginal estimation in real-world problems and the results show improvements against state of the art methods.

Overall the paper seems to develop a solution to a challenging problem. The results are convincing since they compare against several state of the art methods. The use of the SMT framework and guiding it with the weight function seems to be a novel one. Maybe the weakness is the paper is not easy to read particularly for someone who is not an expert in this specific area. I think if the examples are presented more intuitively it can help significantly improve the readability. But overall, it seems to be solving a hard general problem and achieving good results compared to state of the art.

**Q7 Justification For Your Score:**

The combination of using SMT and using the weight function structure seems novel. The results show the gains in comparison to several state of the art methods.

**Q9 Complying With Reviewing Instructions:**

1: Yes.

---

### Official Review · Reviewer_cgwm · 2022-04-17

**Q2(1) Originality/Novelty:** 3
**Q2(2) Significance/Impact:** 3
**Q2(3) Correctness/Technical Quality:** 3
**Q2(6) Clarity Of Writing:** 4
**Q6 Overall Score:** 7
**Q8 Confidence In Your Score:** 2

**Q1 Summary And Contributions:**

This paper focuses on weighted model integration and proposes SA-WMI-PA, an algorithm that combines SMT and problem structure so that it can avoid generating redundant models thus improving the model efficiency. The model is evaluated on two kinds of datasets and has been proved to excel the SOTA baselines.

**Q2 Assessment Of The Paper:**

More detailed information regarding each of these aspects is given below:

**Q2(4) Quality Of Experiments (Optional):**

3: Good: The experimental evaluation is adequate, and the results convincingly support the main claims.

**Q2(5) Reproducibility:**

3: Good: Key resources (e.g., proofs, code, data) are available and key details (e.g., proofs, experimental setup) are sufficiently well-described for competent researchers to confidently reproduce the main results.

**Q3 Main Strengths:**

1. The proposed method is novel and the paper is generally well-written. The experiments are conducted on both synthetic and real-world datasets.

2. This paper gives a deep analysis of existing WMI techniques including the KC for WMI  and WMI-PA.

**Q4 Main Weakness:**

It would be better to give more explanation about the terminology so that the readers not familiar with this area can easily understand the paper.

**Q5 Detailed Comments To The Authors:**

No more.

**Q7 Justification For Your Score:**

The proposed method is novel and efficient.

**Q9 Complying With Reviewing Instructions:**

1: Yes.

---

### Official Review · Reviewer_Bjpr · 2022-04-18

**Q2(1) Originality/Novelty:** 3
**Q2(2) Significance/Impact:** 3
**Q2(3) Correctness/Technical Quality:** 3
**Q2(6) Clarity Of Writing:** 3
**Q6 Overall Score:** 6
**Q8 Confidence In Your Score:** 1

**Q1 Summary And Contributions:**

The paper presents an improvement to weighted model integration. Specifically, the paper builds on the observation that standard techniques do not consider the structure of the weight function and therefore may be performing an excessive amount of avoidable work. The paper alters the technique to make use of this information and provides experimental evaluation showing the benefit of doing so. The technique is able to reduce execution time and solve previously unsolvable problems.

**Q2 Assessment Of The Paper:**

More detailed information regarding each of these aspects is given below:

**Q2(4) Quality Of Experiments (Optional):**

3: Good: The experimental evaluation is adequate, and the results convincingly support the main claims.

**Q2(5) Reproducibility:**

3: Good: Key resources (e.g., proofs, code, data) are available and key details (e.g., proofs, experimental setup) are sufficiently well-described for competent researchers to confidently reproduce the main results.

**Q3 Main Strengths:**

The evaluation reveals the technique proposed to significantly improve the number of problems that can be solved in a reasonable time and the solution times of these problems.

Practical examples nicely illustrate the problems and the presented technique.

**Q4 Main Weakness:**

The experimental results are quite poorly described. The methodology used is not explained, but simply notes a reference to another work. For the second paper, the benchmarks used are presented in the appendix rather than the main paper. In both cases, this interferes with the paper being properly self-contained.

**Q5 Detailed Comments To The Authors:**

The text in figures 2 and 3 is too small to read and both have very long captions. It might be worth considering whether all this is really needed or whether some can be moved to the appendix. It is unusual to see so much explanation in the captions - normally I'd expect to see the discuss about what the figure is showing in the main text.



**Q7 Justification For Your Score:**

The paper makes a very good improvement to a particular problem.

**Q9 Complying With Reviewing Instructions:**

1: Yes.

---

### Decision · Program_Chairs · 2022-05-15

**Decision:**

Accept (Poster)

**Comment:**

Meta Review: The paper contributes a significant advance to the WMI literature along with strongly encouraging empirical results.  Reviewers unanimously recommend acceptance.  While the paper is necessarily dense, it is hoped that the authors can use the camera-ready revision to revise the presentation to address reviewer concerns.